# Long-Term Outcomes and Prognostic Factors of Endoscopic Submucosal Dissection for Early Gastric Cancer in Patients Aged ≥75 Years

**DOI:** 10.3390/cancers12113222

**Published:** 2020-10-31

**Authors:** Jin Won Chang, Da Hyun Jung, Jun Chul Park, Sung Kwan Shin, Sang Kil Lee, Yong Chan Lee

**Affiliations:** Department of Internal Medicine, Yonsei University College of Medicine, 50-1 Yonsei-ro, Seodaemun-gu, Seoul 03722, Korea; hangulchang@yuhs.ac (J.W.C.); JUNGDH@yuhs.ac (D.H.J.); junchul75@yuhs.ac (J.C.P.); KAARMA@yuhs.ac (S.K.S.); LEEYC@yuhs.ac (Y.C.L.)

**Keywords:** endoscopic submucosal dissection, early gastric cancer, elderly, prognostic factor

## Abstract

**Simple Summary:**

The number of elderly patients with early gastric cancer (EGC) who meet the indications for endoscopic submucosal dissection (ESD) is increasing. Since elderly patients have more comorbidities, and lower life expectancy than younger patients, special considerations to establish treatment strategies according to prognosis is needed. We investigate overall survival and risk factors related to survival after ESD in these patients. During the follow-up, the 3-, 5-, and 10-year overall survival was 91.2%, 83.5%, and 54.5%, respectively. Smoking, history of cancer of other organs, neutrophil to lymphocyte ratio > 1.6, Charlson comorbidity index ≥ 3, and presence of lymphovascular invasion were independent five risk factors for poor overall survival. The long-term outcome of ESD was poorer in elderly patients with risk factors than in those without. These prognostic factors can also be useful in deciding whether to recommend additive surgery and a close follow-up after non-curative endoscopic treatment in the elderly patients.

**Abstract:**

Background and Aims: The number of elderly patients with early gastric cancer (EGC) who meet the indications for endoscopic submucosal dissection (ESD) is increasing. We aimed to evaluate the clinical outcomes and prognostic factors of overall survival (OS) in elderly patients undergoing ESD for EGC. Methods: Between January 2006 and December 2018, 439 patients aged ≥75 years who underwent ESD for EGC were analyzed. The clinical outcomes and prognosis were evaluated, and independent risk factors for OS were identified. Results: The mean patient (302 men, 137 women) age was 78.3 (range 75–92) years. En bloc, R0, and curative resections were achieved in 96.8%, 90.7%, and 75.6%, respectively, without severe adverse events. During the follow-up (median 54.2 (range 4.0–159.6) months), 86 patients died (three of gastric cancer). The 3-, 5-, and 10-year OS was 91.2%, 83.5%, and 54.5%, respectively, and the 3-, 5-, and 10-year cancer related survival rate were 99.7%, 99.1% and 97.5%, respectively. In multivariate analysis, smoking, history of cancer of other organs, NLR > 1.6, Charlson comorbidity index ≥ 3, and presence of lymphovascular invasion (hazard ratio = 3.96, 1.78, 1.83, 1.83, and 2.63, respectively, all *p* < 0.05) were independent five risk factors for poor OS. The high-risk group (≥3 risk factors) showed a significantly lower OS than the low-risk group (<2 risk factors) (*p* < 0.001). Conclusions: The five factors could be useful in predicting the long-term prognosis of elderly ESD patients or deciding the therapeutic approaches in case of non-curative resection.

## 1. Introduction

Gastric cancer is one of the major causes of cancer-related deaths worldwide [1] and has the fifth highest incidence among cancers worldwide [2,3,4]. With the aging of the society, the proportion of elderly patients with gastric cancer is increasing [5,6] and the chances of encountering elderly patients with early gastric cancer (EGC) who meet the indications for endoscopic treatment are increasing. Endoscopic submucosal dissection (ESD) is widely accepted as a less invasive treatment compared to surgery and is increasingly performed in elderly patients [7,8,9,10,11,12]. However, the treatment strategy for elderly patients with EGC is not standardized, and data on clinical outcomes in elderly patients with EGC undergoing ESD are limited [13,14] because elderly patients have more comorbidities, and lower life expectancy than younger patients [14].

The cumulative 5-year risk of the progression of EGC to an advanced stage is 63% [15]. Therefore, long-term follow-up data and information on the prognosis of EGC in elderly patients are needed to evaluate the usefulness of ESD for this patient population [14]. In a previous study, elderly patients who underwent non-curative endoscopic resection with additive treatment showed better survival outcomes than those who did not receive additive treatment; thus, additive treatment is recommended in patients undergoing non-curative endoscopic resection without severe comorbidities even if they are ≥75 years old [16]. However, it is challenging for clinicians to decide the therapeutic strategy for EGC in elderly patients because of age-related performance status. Therefore, it is necessary to identify risk factors related to survival after ESD and establish treatment strategies according to prognosis in elderly patients with EGC.

In older patients, there were few studies on short- and long-term outcomes and prognostic factors associated with ESD that have been published. Therefore, we evaluated the clinical outcomes of elderly patients undergoing ESD for EGC and the prognostic factors for overall survival (OS) in the large-scale, long-term cohort data.

## 2. Methods

### 2.1. Patients

We retrospectively reviewed the data of patients aged ≥75 years who underwent ESD at Severance Hospital between January 2006 and December 2018. The data were obtained from a prospectively established cohort database.

The exclusion criteria were as follows: (1) final pathology other than EGC (high-grade dysplasia (n = 34) and other pathologies such as subepithelial tumor, squamous cell carcinoma (n = 25)), (2) insufficient clinical or laboratory information (n = 9), (3) lost to follow-up (n = 17), and (4) impossibility to confirm survival from death (n = 3).

The study protocol was in accordance with the ethical guidelines of the 1975 Declaration of Helsinki and was approved by the Institutional Review Board of Severance Hospital (approval no. 4-2020-0399). The requirement for informed consent was waived owing to the retrospective design of this study.

### 2.2. Evaluation of Baseline Patient Characteristics

We assessed the patient baseline demographic and clinical characteristics, including age, sex, presence of comorbidities, smoking history, alcohol history, and use of anticoagulant or antiplatelet medications. We also evaluated some possible prognostic factors, including the Onodera prognostic nutritional index (PNI) [17], neutrophil-to-lymphocyte ratio (NLR) [18], American Society of Anesthesiologists-performance status (ASA-PS) [19], and Charlson comorbidity index (CCI) [20].

The baseline patient characteristics were assessed using the results of the medical interview, and blood sampling within 2 months before ESD. Medical interviews were recorded through interviews with all patients first hospitalized for ESD. These records were collected prospectively and recorded in a database, and data obtained through follow-up observations were obtained retrospectively. Patient’s alcohol history was recorded quantitatively and smoking history was recorded as whether they are current smoker, past smoker, or non-smoker. CCI was calculated as the sum of scores assigned for several comorbidities based on the original definition [20]. PNI and NLR were calculated based on the results of blood sampling.

### 2.3. Indication and Curability of ESD

The ESD indications and curability for EGC in this study were based on the Japanese Gastric Cancer Treatment Guidelines 2018 (5th edition) [21].

### 2.4. Endoscopic and Histopathologic Evaluation of Lesion Characteristics

The endoscopic and histopathologic evaluation of lesion characteristics in this study were based on the Japanese Gastric Cancer Treatment Guidelines 2018 (5th edition) [21].

### 2.5. Short-Term Outcomes

Short-term clinical outcomes were evaluated in terms of en bloc resection, R0 resection defined as en bloc resection with histologically free lateral and vertical margins [13], curative resection, procedure time, duration of hospital stay for ESD, and adverse events. Procedure time was defined as from the start of dissection for ESD until the dissection is completed and the lesion is separated. Post-ESD bleeding was defined as clinical or laboratory signs of bleeding with confirmation of bleeding on endoscopy. Clinical signs of bleeding include hematemesis or melena or hematochezia, and laboratory sign of bleeding was defined as a ≥2.0 g/dL decrease in hemoglobin level. Perforation was diagnosed endoscopically during the procedure or radiologically through abdominal radiography or computed tomography (CT) after the procedure. Pneumonia was diagnosed with chest radiography or chest CT after the procedure.

### 2.6. Long-Term Outcomes

Long-term follow-up data were retrospectively collected from the medical records at our institution if the patients had regularly visited our institution. The all cause of mortality data of patients who did not regularly visit our institution were obtained from the National Health Insurance Corporation database. The date when EGC was first diagnosed was defined as the index date, and the date of death from the index date was calculated. The metachronous EGC that was additionally detected was ignored. The relationship between OS and clinicopathologic factors was evaluated. The clinicopathologic factors included patient characteristics (sex, smoking history, alcohol history, comorbidities, use of anticoagulants and/or antiplatelet drugs, PNI, NLR, ASA-PS, and CCI), lesion characteristics (tumor size, histologic type, and presence of LVI), en bloc resection, and ESD curability. If more than two lesions were present, lesion characteristics were based on the findings of the most advanced lesion.

### 2.7. Statistical Analysis

The patients’ demographic, pathologic, and short-term clinical outcome data are summarized as the mean (minimum–maximum) for continuous variables and as number with percentages for categorical variables. OS was calculated using the Kaplan–Meir method and compared using the log-rank test. The index date was the date of the patient’s first ESD, and OS was measured from the index date to the date of death or final follow-up or the latest confirmation of survival. The relationship between OS and clinicopathologic features was assessed with univariate analysis using log-rank test. The cutoff values of PNI and NLR were determined as the values that maximized the sum of sensitivity and specificity for OS in receiver operating characteristic curve analysis. The value of dividing high- and low-risk groups was based on the sum of sensitivity and specificity, when the correlation between the patient’s number of risk factors and the OS was analyzed through the receiver operating characteristic curve [22]. The relationships of the patient and lesion characteristics to OS were evaluated using multivariate analysis with a Cox proportional hazard model, and hazard ratios (HRs) and 95% confidence intervals (CI) were calculated.

All statistical analyses were performed using the Statistical Package for the Social Sciences version 25.0 software (IBM Corp., Armonk, NY, USA). A *p* value of <0.05 was considered statistically significant.

## 3. Results

### 3.1. Patient Characteristics

Among the 527 patients (aged ≥75 years) who underwent ESD between January 2006 and December 2018, 88 patients were excluded according to our established exclusion criteria. Hence, a total of 439 patients were selected for the statistical analysis.

The baseline characteristics of the study population are shown in Table 1. The median patient age was 78.3 years (range 75–92 years), and male participants were predominant in this study population (n = 302, 68.8%). The median PNI was 50.1 (range 35.0–115.0) and the NLR was 3.2 (range 0.5–36.0). The ASA-PS score was distributed from 1 to 4, with most patients having a score of 2 (n = 276, 62.9%). The CCI varied between 0 and 6, with most patients (n = 397, 90.4%) having a CCI of ≤3.

### 3.2. Lesion Characteristics

The lesion characteristics of the 531 EGC lesions in the 439 patients are shown in Table 2. Most EGCs (n = 413, 94.1%) occurred in the middle or lower stomach. Most lesions (91.3%) were of the differentiated type. Twenty-three lesions (5.2%) had invaded to the submucosal layer. In most patients (n = 394, 89.8%), the tumor size was ≤30 mm. LVI was observed in 50 (11.4%) patients and perineural involvement was observed in one (0.2%) patient.

### 3.3. Short-Term Outcomes

The short-term clinical outcomes are shown in Table 3. In the 439 patients, the en bloc resection rate was 96.8% (n = 425) and the R0 resection rate was 90.7% (n = 398). Curative resection was achieved in 75.6% (n = 332). The adverse events of ESD were divided into those that occurred before 48 h and those that occurred after 48 h. Adverse events that occurred within 48 h of ESD included post-ESD bleeding (11 (2.5%) cases) and perforation (12 (2.7%) cases). Adverse events that occurred after 48 h of ESD included post-ESD bleeding (10 (2.3%) cases) and pneumonia (11 (2.5%) cases). Each adverse event was treated successfully without requiring surgery. There were no ESD-related deaths. The median procedure time was 30 min (range 5–300 min), and the median duration of hospital stay for ESD was 4 days (2–15 days).

### 3.4. Long-Term Outcomes

During the follow-up period (median 54.2 months, range 4.0–159.6 months), a total of 86 (19.6%) patients died, three (0.7%) of whom died of gastric cancer and 83 (18.9%) of whom died of other causes other than gastric cancer. (Appendix A). Among the 107 (24.4%) patients who had non-curative resection, 72 (67.3%) patients underwent surgery and 35 (32.7%) patients were followed up without additional surgery. For all the patients, the 3-, 5-, and 10-year OS rates were 91.2%, 83.5%, and 54.5%, respectively, and the 3-, 5-, and 10-year cancer related survival rates were 99.7%, 99.1% and 97.5%, respectively (Figure 1). When the patients were divided into the curative resection group and the non-curative resection group, the 3-, 5-, and 10-year OS rates in the curative resection group were 91.0%, 86.9%, and 58.3%, respectively, and those in the non-curative resection group were 91.6%, 72.7%, and 43.8%, respectively, with statistically significant differences (*p* = 0.037) (Appendix A).

When the patients who had non-curative resection were divided into those who received additional resection and those who did not, the 3-, 5-, and 10-year OS rates in the patients underwent additional resection group were 95.4%, 78.3%, and 43.8%, respectively, and the 3-, 5-, and 10-year OS rates in the patients who were followed up without additional resection were 84.3%, 63.0%, and 40.0%, respectively, without statistically significant differences (*p* = 0.310). There was no statistically significant difference in overall survival between the curative resection group and the non-curative resection group who received additional resection (*p* = 0.290), but there was a statistically significant difference in overall survival between the curative resection group and the non-curative resection group who did not receive additional resection (*p* = 0.024).

The relationship between OS and clinicopathologic features is shown in Table 4. In univariate analysis, male sex, smoking, heavy alcohol drinking, history of cancer of other organs, NLR > 1.6, CCI ≥ 3, and presence of LVI were significantly associated with OS. In multivariate analysis, smoking (HR = 3.96; 95% CI, 2.05–7.65; *p* < 0.001), history of cancer of other organs (HR = 1.78; 95% CI, 1.30–4.52; *p* = 0.006), NLR >1.6 (HR = 1.83; 95% CI, 1.04–3.21; *p* = 0.035), CCI ≥ 3 (HR = 1.83; 95% CI, 1.32–3.20; *p* = 0.001), and presence of LVI (HR = 2.63; 95% CI, 1.21–5.73; *p* = 0.015) were independent risk factors for poor OS (Table 5). When calculating how many of these five risk factors patients have, the correlation with the OS through the receiver operating characteristic curve, the sum of sensitivity and specificity is the highest when divided by three. The presence of three or more of these five risk factors were defined as “high risk”; accordingly, 328 (74.7%) patients were in the low-risk group and 111 (25.3%) patients were in the high-risk group. When comparing the two groups in terms of OS, the high-risk group showed a significantly lower OS than the low-risk group (*p* < 0.001). In the low-risk group, the 3-, 5-, and 10-year OS rates were 94.6%, 90.1%, and 63.8%, respectively. In the high-risk group, the 3-, 5-, and 10-year OS rates were 80.0%, 62.5%, and 22.1%, respectively (Appendix A).

## 4. Discussion and Conclusions

The treatment strategy for elderly patients has become an important issue. In the Republic of Korea (ROK), the average life expectancy in 2017 was 82.7 years [23]; thus, how to treat EGC in elderly patients aged ≥75 years is an important issue. In elderly patients, a different approach from that used in younger patients for the treatment of EGC may be needed because of the comorbidities and limited life expectancy related to aging. However, the long-term outcomes and prognostic factors in elderly patients with EGC have been evaluated in only a few studies [13,24,25,26]. In order to analyze the OS after ESD in elderly patients, a long observation period is mandatory in sufficient number of patients. The fact that our study had an average observation period of 54.2 months (range 4.0–159.6 months) in 439 patients can be considered as a significant advantage compared to other studies. Therefore, we analyzed the long-term outcomes and prognostic factors in elderly patients who underwent ESD for EGC.

Compared with surgery, ESD is less invasive and improves the quality of life of patients [11]. With improvements in ESD devices and procedural techniques, performing ESD in patients with comorbidities has been shown to be safe and feasible [27]. However, care should be taken when deciding ESD for EGC in elderly patients because of their poor physical status and life expectancy. Therefore, creating a decision algorithm considering the long-term outcomes and prognostic factors of ESD for EGC in elderly patients may be necessary.

We identified the short- and long-term outcomes and prognostic factors of patients aged ≥75 years who underwent ESD for EGC. The short-term outcomes of en bloc resection (96.8% vs. 92.7–98%), R0 resection (90.7% vs. 91.7–93.0%), curative resection (75.6% vs. 73.6%), perforation (2.7% vs. 1.8–6.0%), and delayed bleeding (2.3% vs. 4.0–4.5%) were similar in general patients studied in previous studies [7,9,10,28] and in the elderly patients in this study. There were no treatment-related deaths and no severe adverse events that required surgery. These results indicate that ESD can be safely performed even in patients aged ≥75 years. Therefore, if the patient meets the indications for ESD, the procedure should be considered even if the patient is elderly.

When patients have undergone a non-curative resection that requires additional surgery after ESD, their general condition should be carefully assessed considering the adverse events associated with general anesthesia and surgery. As the results of the previous paper published by our institution have also been revealed in this study, patients who have not undergone additional resection after non-curative resection had a difference in overall survival compared to patients who received complete resection. Therefore, additional surgery after ESD should be discussed. Eighty-three (18.9%) patients in this study died of non-gastric cancer causes during the follow-up period, 7.5% died within 3 years and 11.8% died within 5 years after ESD. If patients die shortly after the diagnosis of EGC, it may not be reasonable to perform ESD or surgery after non-curative ESD. Therefore, it is important to identify the prognostic factors and determine the treatment strategy associated with OS.

In our study, smoking, history of other cancers, NLR > 1.6, CCI ≥ 3, and presence of LVI were the independent prognostic factors in elderly EGC patients undergoing ESD. In a previous study, CCI ≥ 3 was identified as a prognostic factor in elderly patients aged ≥75 years with EGC after non-curative ESD [26]. PNI was identified as a prognostic factor in a previous study in patients aged ≥85 years; however, in our study, PNI was not a prognostic factor of ESD for EGC in elderly patients. As PNI was developed to predict prognosis in advanced gastrointestinal cancer [17], the results may have been different in this EGC cohort. The results of this study showed statistically significant differences in the 3-, 5-, and 10-year OS between the low-risk and high-risk groups, which may be helpful in deciding whether to perform additional surgery after non-curative ESD by examining the number of risk factors a patient has and whether the patient belongs to the high-risk or low-risk group.

Our study has several clinical implications. First, to our knowledge, our study included and analyzed the largest sample size of patients aged ≥75 years who underwent ESD for EGC. Our study attempted to secure statistical reliability by including a larger sample size than that in previous studies. Additionally, the sufficient number of mortality cases (n = 86, 19.6%) during the follow-up period might support the reliability of our study. Long-term prognosis and survival analyses were possible because the follow-up period was long and the ratio of mortality was not small. Second, considering that the patients analyzed in this study were aged ≥75 years, the median follow-up of 4.5 years (maximum 13.3 years) was sufficient to identify long-term outcomes. This study showed the long-term prognosis of ESD in elderly patients and may serve as a basis for the determination in real clinical practice and for further research. Third, through the risk factors identified in this study, we were able to classify patients into low-risk and high-risk groups of OS after ESD. From these results, we provided data that can be used as a basis for judgment when there is a concern about whether to perform ESD in elderly patients.

Our study also has some limitations. First, despite the fact that our study analyzed data which were prospectively collected, our study might be subject to a potential bias because of its retrospective nature. The data that are not included in the initial clinical protocol are bound to be limited in interpretation or analysis. Further prospective studies on the prognosis and risk factors of ESD in the elderly are needed to validate the study results. Second, this was a single-center study conducted in the ROK. As the patients included in this study were from our institution, which is an academic teaching hospital in the ROK, our patients may not represent the entire elderly population. Therefore, generalizing our results to the entire population or applying them to Western patients may be difficult. Further multinational and multicenter large cohort studies should be conducted to validate the study results. Third, this study did not include patients who did not undergo ESD and were only followed up after the diagnosis of EGC. For a more accurate comparison of clinical outcomes, it is necessary to compare the group of patients who had undergone ESD after being diagnosed with EGC and those who were observed without undergoing ESD. Fourth, the period from the beginning to the end of the study was about 13 years. During the study period, there have been advances in techniques and devices, which may have caused a difference in prognosis and outcomes. The technology of ESD is evolving gradually and has made endoscopic resection safer, easier, and more reliable. Many types of knives including IT knife, hook knife, flex knife, dual knife, hybrid knife, etc., have been developed and technologically advanced [29]. In addition, various accessories, hemostasis methods, submucosal injection solutions have been developed, and the experience of the operator must have been accumulated. In this study, we did sub-analysis of techniques and devices on OS, but we could not get find significant difference. This may be because ‘techniques and devices’ are mixtures of various factors such as knife type, hemostasis method, difference in submucosal injection solution, and skill level of the operator.

In conclusion, we identified that smoking, history of cancer of other organs, NLR > 1.6, CCI ≥ 3, and presence of LVI were the independent risk factors for poor OS in elderly patients undergoing ESD for EGC. The long-term outcome of ESD was poorer in elderly patients with comorbidities than in those without. Thus, the abovementioned prognostic factors can be used to decide the therapeutic approach for EGC in elderly patients, including the necessity of additive surgery and a close follow-up after non-curative ESD for EGC.

## Figures and Tables

**Figure 1 cancers-12-03222-f001:**
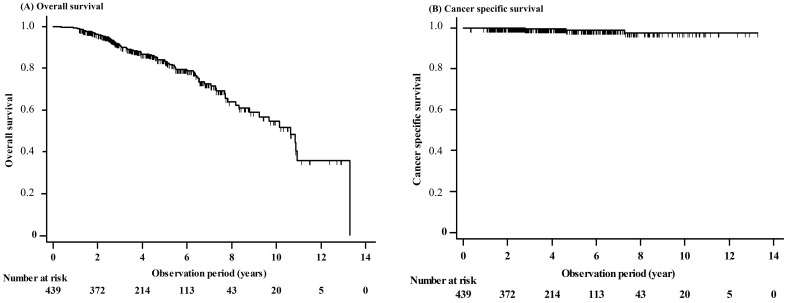
Relationship between OS and clinicopathologic factors. Kaplan–Meier estimation of overall survival (OS) in 439 elderly patients who underwent ESD for EGC. (**A**) The 3-, 5-, and 10-year OS rates were 91.2%, 83.5%, and 54.5%, respectively, and (**B**) the 3-, 5-, and 10-year cancer related survival rate were 99.7%, 99.1% and 97.5%, respectively. ESD, endoscopic submucosal dissection; EGC, early gastric cancer.

**Table 1 cancers-12-03222-t001:** Baseline characteristics of the 439 patients aged ≥75 years received endoscopic submucosal dissection (ESD) for early gastric cancer (EGC).

Variables	Value
Demographic variables	
Age, years (range)	78.3 (75–92)
Male gender	302 (68.8)
Smoking history	219 (49.9)
Heavy alcoholics	36 (8.2)
Comorbidities (with overlap)	
Cardiovascular disease	74 (16.9)
Cerebrovascular disease	24 (5.5)
Cancer of the other organs	53 (12.1)
Kidney disease	18 (4.1)
Diabetes mellitus	101 (23.0)
Hypertension	255 (58.1)
Use of anticoagulants and/or antiplatelet drugs	143 (32.6)
Prognostic factors	
Prognostic nutritional index (range)	50.1 (35.0–115)
Neutrophil to lymphocyte ratio (range)	3.2 (0.5–36.0)
ASA-PS score	
1	21 (4.8)
2	276 (62.9)
3	135 (30.8)
4	7 (1.6)
Charlson comorbidity index	
1	1 (0.2)
2	260 (59.2)
3	136 (31.0)
4	33 (7.5)
5	7 (1.6)
6	2 (0.5)

Variables are expressed as mean (minimum-maximum) or n (%). ESD, endoscopic submucosal dissection; EGC, early gastric cancer; ASA-PS, American society of anesthesiologist physical status.

**Table 2 cancers-12-03222-t002:** Characteristics of the 531 EGCs in the 439 patients aged ≥75 years treated by ESD.

Variables	Value
Location	
Upper	41 (9.3)
Middle	100 (22.8)
Lower	313 (71.3)
Remnant stomach	3 (0.7)
Number of lesions	
1	362 (82.5)
2	64 (14.6)
3	11 (2.5)
4	2 (0.5)
Macroscopic type	
Elevated	109 (24.8)
Flat	46 (10.5)
Depressed	44 (10.0)
Mixed	240 (54.7)
Tumor size (mm), median	
Tumor size	
≤20 mm	326 (74.3)
21–30 mm	68 (15.5)
>30 mm	45 (10.2)
Tumor depth	
Mucosa	416 (94.8)
Submucosa	23 (5.2)
Histologic type	
Differentiated	401 (91.3)
Undifferentiated	38 (8.7)
Lymphovascular involvement	
Present	50 (11.4)
Absent	389 (88.6)
Perineural involvement	
Present	1 (0.2)
Absent	438 (99.8)

Variables are expressed as mean (minimum-maximum) or n (%). ESD, endoscopic submucosal dissection; EGC, early gastric cancer.

**Table 3 cancers-12-03222-t003:** Short-term clinical outcomes of endoscopic submucosal dissection for the 439 elderly patients.

Variables	Value
En bloc resection	425 (96.8)
R0 resection	398 (90.7)
Curative resection	332 (75.6)
Adverse event (<48 hrs)	
Bleeding	11 (2.5)
Perforation	12 (2.7)
Pneumonia	0 (0.0)
Adverse event (>48 hrs)	
Bleeding	10 (2.3)
Perforation	0 (0.0)
Pneumonia	11 (2.5)
Procedure time (minutes), median (range)	30 (5–300)
Duration of hospital stay for ESD (days), median (range)	4 (2–15)

ESD, endoscopic submucosal dissection.

**Table 4 cancers-12-03222-t004:** Relationship between overall survival and clinicopathologic factors.

Clinicopathologic Features	Number of Patients	3-Year OS, % (95% CI)	5-Year OS, % (95% CI)	10-Year OS, % (95% CI)	*p*-Value
Patients characteristics					
Sex					0.025
Male	302	88.8 (84.9–92.7)	82.1 (77.0–87.2)	48.1 (34.4–61.8)	
Female	137	96.6 (93.3–99.9)	87.2 (79.9–94.5)	66.2 (51.3–81.1)	
Smoking					<0.001
Smoker	219	87.6 (82.9–92.3)	78.2 (71.5–84.9)	42.6 (26.3–58.9)	
Non-smoker	220	94.6 (91.3–97.9)	88.8 (83.5–94.1)	65.0 (51.7–78.3)	
Alcohol					0.006
Heavy	36	79.2 (64.3–94.1)	53.3 (20.6–86.0)	53.3 (20.6–86.0)	
Non-alcohol, social	403	92.1 (89.2–95.0)	84.7 (80.4–89.0)	55.5 (44.9–66.1)	
Cardiovascular disease					0.054
Yes	74	91.4 (84.0–98.8)	76.4 (62.7–90.1)	27.5 (0.3–54.7)	
No	365	91.1 (88.0–94.2)	84.6 (80.1–89.1)	58.0 (46.8–69.2)	
Cerebrovascular disease					0.803
Yes	24	85.4 (70.1–100)	78.3 (58.9–97.7)	78.3 (58.9–97.7)	
No	415	91.5 (88.6–94.4)	83.8 (79.5–88.1)	53.9 (43.1–64.7)	
Cancer of the other organs					0.022
Yes	53	80.7 (68.4–93.0)	72.4 (56.7–88.1)	47.3 (24.2–70.4)	
No	386	92.4 (89.5–95.3)	84.8 (80.5–89.1)	55.6 (44.2–67.0)	
Kidney disease					0.591
Yes	18	75.9 (55.1–96.7)	75.9 (55.1–96.7)	75.9 (55.1–96.7)	
No	421	91.6 (88.7–94.5)	83.9 (79.6–88.2)	54.2 (43.6–64.8)	
DM					0.195
Yes	101	92.8 (89.9–95.7)	84.8(80.1–89.5)	51.3 (31.5–71.1)	
No	338	85.2 (77.4–93.0)	79.3 (69.5–89.1)	55.3 (43.0–67.6)	
HTN					0.749
Yes	255	91.7 (88.0–95.4)	85.6 (80.3–90.9)	50.7 (36.2–65.2)	
No	184	90.3 (85.6–95.0)	80.6 (73.5–87.7)	59.7 (45.2–74.2)	
Use of anticoagulants and/or antiplatelet drugs					0.449
Yes	143	88.0 (81.9–94.1)	81.6 (73.8–89.4)	55.6 (35.0–76.2)	
No	296	92.7 (89.6–95.8)	84.5 (79.4–89.6)	54.7 (42.7–66.7)	
PNI					0.278
≤52.4	306	90.9 (87.4–94.4)	81.4 (75.9–86.9)	50.9 (33.8–68.0)	
>52.4	133	91.9 (86.8–97.0)	88.1 (81.6–94.6)	58.8 (44.1–73.5)	
NLR					0.006
≤1.6	124	94.0 (89.3–98.7)	89.5 (82.8–96.2)	69.3 (54.0–84.6)	
>1.6	315	90.0 (86.5–93.5)	81.2 (75.9–86.5)	47.5 (34.2–60.8)	
ASA-PS					0.753
1 & 2	297	90.6 (87.1–94.1)	83.5 (78.4–88.6)	54.1 (42.9–65.3)	
3 & 4	142	92.5 (87.8–97.2)	83.4 (75.2–91.6)	49.0 (8.8–89.2)	
Charson comorbidity index					0.002
≤2	261	92.7 (89.2–96.2)	87.1 (82.0–92.2)	64.9 (51.0–78.8)	
≥3	178	88.0 (82.5–93.5)	78.2 (70.8–85.6)	41.2 (26.1–56.3)	
Lesion characteristics					
Tumor size					0.068
≤30	394	90.5 (87.4–93.6)	82.4 (77.9–86.9)	52.7 (41.9–63.5)	
>30	45	97.7 (93.4–100)	88.8 (71.7–100)	88.8 (71.7–100)	
Histologic type					0.339
Differentiated	401	91.9 (89.0–94.8)	84.2 (79.7–88.7)	56.3 (54.3–67.3)	
Undifferentiated	38	83.0 (70.5–95.5)	75.9 (61.2–90.6)	48.2 (21.3–75.1)	
Lymphovascular involvement					<0.001
Present	50	86.2 (75.8–86.6)	56.4 (37.2–75.6)	32.3 (2.7–61.9)	
Absent	359	91.6 (88.5–94.7)	86.8 (82.7–90.9)	56.9 (45.5–68.3)	
ESD curability					0.037
Curative	332	91.0 (87.7–94.3)	86.9 (82.6–91.2)	58.3 (45.4–71.2)	
Noncurative	107	91.6 (85.9–97.3)	72.7 (61.5–83.9)	43.8 (26.2–61.4)	
En bloc resection					0.481
En bloc resection	425	91.3 (88.4–94.2)	83.7 (79.4–88.0)	54.4 (43.2–65.6)	
Piecemeal	14	85.7 (67.3–100)	76.2 (52.3–100)	53.3 (21.4–85.2)	

DM, diabetes mellitus; HTN, hypertension; PNI, prognostic nutritional index; NLR, neutrophil to lymphocyte ratio; ASA-PS, American society of anesthesiologist physical status; ESD, endoscopic submucosal dissection.

**Table 5 cancers-12-03222-t005:** Multivariate analysis of factors associated with overall survival.

Variable	HR	95% CI	*p*-Value
Male gender	0.64	0.313–1.298	0.214
Smoker	3.96	2.050–7.652	<0.001
Heavy alcoholics	1.78	0.850–3.740	0.126
Cancer of the other organs	2.42	1.296–4.520	0.006
Neutrophil to lymphocyte ratio >1.6	1.83	1.043–3.211	0.035
Charlson comorbidity index ≥3	2.05	1.317–3.195	0.001
Lymphovascular involvement	2.63	1.206–5.732	0.015
Noncurative resection	1.42	0.746–2.717	0.283

HR, hazard ratio; CI, confidence interval; NLR, Neutrophil to lymphocyte ratio.

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
