# Peer review of "Long-Term Outcomes and Prognostic Factors of Endoscopic Submucosal Dissection for Early Gastric Cancer in Patients Aged ≥75 Years"

_cancers, 2020, doi:10.3390/cancers12113222_

Round 1
Reviewer 1 Report
In My opinion, now the study is more complete, useful, enjoiable, convincing. I can only suggest authors to make abstract more agressive and intriguing to encurage more readers to be interested on this data
Author Response
Response) Thank you for the kind comment. We modified abstract.
Background and Aims: The number of elderly patients with early gastric cancer (EGC) who meet the indications for endoscopic submucosal dissection (ESD) are increasing. We aimed to evaluate the clinical outcomes and prognostic factors of overall survival (OS) in elderly patients undergoing ESD for EGC.
Methods: Between January 2006 and December 2018, 439 patients aged ≥75 years who underwent ESD for EGC were analyzed. The clinical outcomes and prognosis were evaluated, and independent risk factors for OS were identified.
Results: The mean patient (302 men, 137 women) age was 78.3 (range 75–92) years. En bloc, R0, and curative resections were achieved in 96.8%, 90.7%, and 75.6%, respectively, without severe adverse events. During the follow-up (median 54.2 [range 4.0–159.6] months), 86 patients died (3 of gastric cancer). The 3-, 5-, and 10-year OS was 91.2%, 83.5%, and 54.5%, respectively, and the 3-, 5-, and 10-year cancer related survival rate were 99.7%, 99.1% and 97.5%, respectively. In multivariate analysis, smoking, history of cancer of other organs, NLR >1.6, Charlson comorbidity index ≥3, and presence of lymphovascular invasion (hazard ratio = 3.96, 1.78, 1.83, 1.83, and 2.63, respectively, all P<0.05) were independent five risk factors for poor OS. The high-risk group (≥ 3 risk factors) showed a significantly lower OS than the low-risk group (< 2 risk factors) (P<0.001).
Conclusions: The five factors could be useful in predicting the long-term prognosis of elderly ESD patients or deciding the therapeutic approaches in case of non-curative resection.

Reviewer 2 Report
After performing many reviews of this manuscript, I believe the authors suitably addressed the points.
I have no more questions for the authors and it is a much better paper now.
Author Response
We revised the manuscript based on your comments and thank you for helping us make it a better paper. Thank you again.

This manuscript is a resubmission of an earlier submission. The following is a list of the peer review reports and author responses from that submission.
Round 1
Reviewer 1 Report
The study : Long-term outcomes and prognostic factors of endoscopic submucosal dissection for early gastric cancer in patients aged ≥75 years, from Jin Won Chang et al, is an interesting study with a great number of followed patients for a median time of 5 years, treated at a single, but expert institution.
The same group presented a similar study with a lower number of patients; this study regard survival in elderly patients treated with not curative treatment submitted or not to a second procedure; it could be interesting to integrate this study with that one.
I mean that it would be interesting remove some overlapping data (just presented in tables or considered as standard procedure), add information on the different survival rates between patients curativelly treated re treated or not curatively treated and differentiate survival as overal and cancer related; For example, not curativelly treated patients were died for cancer or not ? ( if only 3 patients are died for cancer this coudn't be significant)
Major points
-Line219 : only 3 patients died for gastric cancer. This is different from line 275... 20% died for other causes. This must be better explained
-Table 1 : it is difficult to believe that a patient with a T1a not curativelly treated, will die for gastric cancer after 88 month; unless the authors present a low disease free survival and explain why they consider this correlation
-Fig 1 can be removed. In fig 2, authors can add survival rates differentiated as overal survival and cancer related
Minor points
-Line 107 Authors may clarify why they differentiate heavy alcholist and not heavy smokers
-Line 111 Authors must must describe in ,aterioal the difference between high and low risk pazients ( >3 < ) removing this presentation from discussion and specifying why they choose 3 ( reference?)
-Line 119 Authors may remove from materials, the description of absolute or expanded criteria manteining the reference(The ESD indications for EGC in this study were based on the Japanese Gastric Cancer Treatment Guidelines 2018 (5th edition) [21] )
-Line 135-139 can be removed. Endoscopic and histopathologic evaluation of lesion characteristics can be only referred to japanese guidelines
-Line127-132 can be removed because described in reference 21
-Line 156 authors must define that they consider any cause of death
-from line 186 to 196 authors may remove some simple data just presented in table 1
-line199 the same with lesion characteristics
-line225 remove ...the..
-Discussion can be better sintetized adding consideration on cancer related survival and not curative resection, Patients submitted to re resection had better prognosis also in this setting?
Author Response
Response to reviewers’ comments
We were most delighted to learn that our manuscript entitled “Long-term outcomes and prognostic factors of endoscopic submucosal dissection for early gastric cancer in patients aged ≥75 years” has been given the opportunity to be revised for publication in the Cancers. We have carefully considered the valuable comments and suggestions provided by reviewers and the editor, and made great efforts to improve our paper accordingly. Below are our point-by-point answers to specific questions raised by the reviewers (modifications in the manuscript are underlined in red, bold letters). We hope that the revised version of the manuscript is now deemed suitable for publication.
Reviewer #1
The study: Long-term outcomes and prognostic factors of endoscopic submucosal dissection for early gastric cancer in patients aged ≥75 years, from Jin Won Chang et al, is an interesting study with a great number of followed patients for a median time of 5 years, treated at a single, but expert institution.
The same group presented a similar study with a lower number of patients; this study regard survival in elderly patients treated with not curative treatment submitted or not to a second procedure; it could be interesting to integrate this study with that one.
I mean that it would be interesting remove some overlapping data (just presented in tables or considered as standard procedure), add information on the different survival rates between patients curatively treated re treated or not curatively treated and differentiate survival as overall and cancer related; For example, not curativelly treated patients were died for cancer or not? (if only 3 patients are died for cancer this coudn't be significant)
Major points
-Line219: only 3 patients died for gastric cancer. This is different from line 275... 20% died for other causes. This must be better explained
Response) In this study, the total number of deaths was 86(19.6%), with only three (0.7%) patients died of gastric cancer and 83(18.9%) patients died from other causes. We used “About 20%” for this reason in main text. We modified sentences in discussion like below.
-Table 1: it is difficult to believe that a patient with a T1a not curatively treated, will die for gastric cancer after 88 months; unless the authors present a low disease free survival and explain why they consider this correlation
Response) Thank you for the kind comment. We reviewed the medical record again to confirm this patient. As a result of checking again, it is true that the patient was found to have metastatic gastric cancer during a follow-up and died from it. We think this patient was an unusual case. Case reports similar to this are sometimes reported. [1]
-Fig 1 can be removed. In fig 2, authors can add survival rates differentiated as overall survival and cancer related
Response) Thank you for the kind comment. As mentioned, we removed Figure 1. Figure 2 also calculated cancer-related survival as you gave us the comment. For all the patients, the 3-, 5-, and 10-year OS rates were 91.2%, 83.5%, and 54.5%, respectively, and the 3-,5-, and 10-year cancer related survival rate were 99.7%, 99.1% and 97.5%, respectively. We revised it accordingly.
Minor points
-Line 107 Authors may clarify why they differentiate heavy alcholist and not heavy smokers
Response) Thank you for the detail comment. The patient’s smoking history was recorded only as whether they are current smoker, past smoker, or non-smoker. On the other hand, alcohol history was recorded as the amount of alcohol consumption, so it was possible to analyze whether the patient was heavy alcoholics. We plan to include quantitative data on smoking in the data in the future. We added this information in Results section briefly.
-Line 111 Authors must must describe in,aterioal the difference between high and low risk pazients ( >3 < ) removing this presentation from discussion and specifying why they choose 3 ( reference?)
Response) The value of dividing high and low risk groups was based on the sum of sensitivity and specificity, when the correlation between the patient’s number of risk factors and the OS was analyzed through the ROC curve. We added this information in Methods and Results section briefly [2].
-Line 119 Authors may remove from materials, the description of absolute or expanded criteria manteining the reference(The ESD indications for EGC in this study were based on the Japanese Gastric Cancer Treatment Guidelines 2018 (5th edition) [21] )
Response) We corrected it
-Line 135-139 can be removed. Endoscopic and histopathologic evaluation of lesion characteristics can be only referred to Japanese guidelines
Response) We removed it according to your advice
-Line127-132 can be removed because described in reference 21
Response) We removed it according to your advice
-Line 156 authors must define that they consider any cause of death
Response) We modified sentences to provide correct information.
-from line 186 to 196 authors may remove some simple data just presented in table 1
Response) Thank you for the kind comment. As mentioned, we removed some parts.
-line199 the same with lesion characteristics
Response) We corrected it
-line225 remove ...the..
Response) We corrected it
-Discussion can be better sintetized adding consideration on cancer related survival and not curative resection, Patients submitted to re resection had better prognosis also in this setting?
Response) Thank you for the kind comment. Additional analyzes were performed on non-curative resection patients, and details were added to the Results section.
- Kawabata, H.; Kawakatsu, Y.; Yamaguchi, K.; Ueda, Y.; Okazaki, Y.; Hitomi, M.; Miyata, M.; Motoi, S.; Enoki, Y.; Minamikawa, S. A rare case of local recurrence following curative endoscopic submucosal dissection of intramucosal differentiated-type gastric cancer. Gastroenterology Res 2019, 12, 103-106.
- Habibzadeh, F.; Habibzadeh, P.; Yadollahie, M. On determining the most appropriate test cut-off value: The case of tests with continuous results. Biochem Med (Zagreb) 2016, 26, 297-307.

Reviewer 2 Report
Comments:
The authors analyzed long-term outcomes and prognostic factors of the patient with early gastric cancer treated by the ESD technique. This retrospective study has a large sample size and focused on an interesting theme. However, I would like to see addressed for further improvement of this manuscript.
Major Comments:
Please consider making more clear what was the novelty of this large-scale study.
As the authors described, there have been advances in techniques and devices. I would like to recommend to try sub-analysis or discuss deeper how the differences affect the results.
METHODS
Please consider describing clearly how the authors dealt with the patients who were found metachronous EGC. If the authors define the first EGC as index treatment during study periods, and the duration until detection of secondary found EGC (or advanced gastric cancer, and third or more detected gastric cancer was ignored) was defined as the endpoint. please consider to described so.
RESULTS:
I would like to recommend to show the follow-up rate. One of the important points may be plotting the censoring in the all Kaplan-Meier curve. Censoring plots in all Kaplan-Meier curves are informative.
DISCUSSION:
The authors described that the abovementioned prognostic factors can be used to decide the therapeutic approach for EGC in elderly patients, including the necessity of additive surgery and a close follow-up after non-curative ESD for EGC.
It was difficult to find the data about the additive surgery, therefore, this sentence seemed to have a logical leap.
Minor Comments:
INTRODUCTION:
The authors described that previously some studies on long-term outcomes has been published, “therefore”, the authors evaluated the prognostic factors for OS.
It was difficult to understand why the authors “therefore” evaluated the prognostic factors. It seems to be already published all data. No novelty was found.
METHODS:
The authors described that they excluded “carcinoma in situ”. Did it mean squamous cell carcinoma in situ?
How many participants were excluded because of insufficient clinical laboratory information? Additionally, please consider presenting the patients excluded in figure 1.
The authors described they used the results of the medical interview. At the same time, this study was designed retrospectively. How detailed was the interview? If the medical interview had some missing data due to the retrospective setting, patients included the study was limited.
The authors analyzed using the Cox proportional hazard model. It was the correct method, however, when the authors include the noncurative resection as a covariance, please it might be better to include whether the additional surgery was performed or not. If the number of additional surgery was too small to analyze statistically, please mention and present the information. I could not found data on how many patients were treated by surgery or resected additional treatment.
Please consider describing the number of the excluded patient due to follow-up loss and not possible to confirm survival or death, respectively. It might be 29 in total, but how many in each?
I am wondering the Charlson comorbidity index has an important role in this cohort despite almost all of the patients were include CCI 2 or 3. Did it mean that the difference in CCI 2 or 3 affected the decision making?
The authors analyzed the histologic type dividing into two types, differentiated and undifferentiated. How many mixed type EGCs were there?
There was at least the patient who was treated within 5 minutes by ESD technique. What was the definition of the procedure time?
DISCUSSION:
The authors described that it may not be reasonable to perform ESD or surgery after non-curative ESD. I would like to agree with this opinion, however, I could not find the data on how many patients treated by additional surgery or rejected additional treatment. It was difficult to understand this sentence without data on additional surgery.
The authors described that the results of this study showed statistically significant differences in the 4-, 5-, and 10-year OS between the low-risk and high-risk groups (P21 L288).
It was difficult to find the results that showed significant differences.
Author Response
Response to reviewers’ comments
We were most delighted to learn that our manuscript entitled “Long-term outcomes and prognostic factors of endoscopic submucosal dissection for early gastric cancer in patients aged ≥75 years” has been given the opportunity to be revised for publication in the Cancers. We have carefully considered the valuable comments and suggestions provided by reviewers and the editor, and made great efforts to improve our paper accordingly. Below are our point-by-point answers to specific questions raised by the reviewers (modifications in the manuscript are underlined in red, bold letters). We hope that the revised version of the manuscript is now deemed suitable for publication.
Reviewer #2
Comments: The authors analyzed long-term outcomes and prognostic factors of the patient with early gastric cancer treated by the ESD technique. This retrospective study has a large sample size and focused on an interesting theme. However, I would like to see addressed for further improvement of this manuscript.
Major Comments:
Please consider making more clear what was the novelty of this large-scale study.
Response) Thank you for the kind comment. There are several papers related to ESD, but there are not many papers that have analyzed issues of ESD for elderly patients. The average age of the patients included in our study was 78.3 years (range 75–92 years), and relatively older patients were included compared to other studies. In order to analyze the OS after ESD in elderly patients, a long observation period is mandatory in sufficient number of patients. The fact that our study had an average observation period of 54.2 months (range 4.0–159.6 months) in 439 patients can be considered as a significant advantage compared to other studies. As the reviewer pointed out, we added the strengths of our study to the discussion section.
As the authors described, there have been advances in techniques and devices. I would like to recommend to try sub-analysis or discuss deeper how the differences affect the results.
Response) Thank you for the kind comment. We have described additional details on techniques and devices as pointed out by the reviewer. And, sub-analysis of techniques and devices was attempted, but we could not find significant difference. This may be because ‘techniques and devices’ are mixtures of various factors such as knife type, hemostasis method, difference in submucosal injection solution, and skill level of the operator.
METHODS
Please consider describing clearly how the authors dealt with the patients who were found metachronous EGC. If the authors define the first EGC as index treatment during study periods, and the duration until detection of secondary found EGC (or advanced gastric cancer, and third or more detected gastric cancer was ignored) was defined as the endpoint. please consider to described so.
Response) In this study, the date when EGC was first diagnosed was defined as the index date, and the date of death from the index date was calculated and used for OS analysis. Further analysis was not performed on metachronous EGC after the first diagnosis. However, when analyzing the patients who died, there were no cases of death from the metachronous cancer, so it is not expected to have a significant effect on the prognosis. We added this information in the Methods section.
RESULTS:
I would like to recommend to show the follow-up rate. One of the important points may be plotting the censoring in the all Kaplan-Meier curve. Censoring plots in all Kaplan-Meier curves are informative.
Response) Thank you for the good points. As you mentioned we modified the Figure.
DISCUSSION:
The authors described that the above mentioned prognostic factors can be used to decide the therapeutic approach for EGC in elderly patients, including the necessity of additive surgery and a close follow-up after non-curative ESD for EGC.
It was difficult to find the data about the additive surgery, therefore, this sentence seemed to have a logical leap.
Response) Thank you for the kind comment. It seems that there is a logical error in the part you mentioned. In order to eliminate misunderstanding, the sentence of the Discussion section was modified.
Minor Comments:
INTRODUCTION:
The authors described that previously some studies on long-term outcomes has been published, “therefore”, the authors evaluated the prognostic factors for OS.
It was difficult to understand why the authors “therefore” evaluated the prognostic factors. It seems to be already published all data. No novelty was found.
Response) Thank you very much for pointing out our fault.
In older patients, there were few studies on short- and long-term outcomes and prognostic factors associated with ESD have been published. Therefore, we evaluated the clinical outcomes of elderly patients undergoing ESD for EGC and the prognostic factors for overall survival (OS) in the large-scale, long-term cohort data. We modified sentences to provide correct information in the Introduction section
METHODS:
The authors described that they excluded “carcinoma in situ”. Did it mean squamous cell carcinoma in situ?
Response) Thank you very much for pointing out our fault. Actually we don’t have case with squamous cell carcinoma in situ, we delete those words.
How many participants were excluded because of insufficient clinical laboratory information? Additionally, please consider presenting the patients excluded in figure 1.
Response) Thank you for the kind comment. Nine patients were excluded for lack of data. As you mentioned we added this information.
The authors described they used the results of the medical interview. At the same time, this study was designed retrospectively. How detailed was the interview? If the medical interview had some missing data due to the retrospective setting, patients included the study was limited.
Response) To give an overview of our study, it is a retrospective analysis study using data prospectively collected for ESD patients. In our institution, the patient’s list and data are prospectively collected by specialized nurses for all patients undergoing ESD using established clinical protocols. Therefore, as pointed out by the reviewer, data that are not included in the initial clinical protocol are bound to be limited in interpretation or analysis. For this, the methods and discussion were supplemented.
The authors analyzed using the Cox proportional hazard model. It was the correct method, however, when the authors include the noncurative resection as a covariance, please it might be better to include whether the additional surgery was performed or not. If the number of additional surgery was too small to analyze statistically, please mention and present the information. I could not found data on how many patients were treated by surgery or resected additional treatment.
Response) Thank you for the informative comment. We did further analysis as reviewer recommended.
Among the 107 (24.4%) patients who had non-curative resection, 72 (67.3%) patients underwent surgery and 35 (32.7%) patients were followed up without additional surgery. Additional analysis was performed and there was no statistically significant difference in overall survival between the curative resection group and the non-curative resection group who received additional resection (P=0.290), but there was a statistically significant difference in overall survival between the curative resection group and the non-curative resection group who did not receive additional resection (P=0.024). The above analysis results were the same as the paper previously reported by our institution. We added this information at Result and Discussion section.
Please consider describing the number of the excluded patient due to follow-up loss and not possible to confirm survival or death, respectively. It might be 29 in total, but how many in each?
Response) Among the 29 excluded patient, 9 were excluded due to insufficient clinical or laboratory information, 17 were follow-up loss and 3 were excluded due to impossibility to confirm survival from death. We added this information at Methods section.
I am wondering the Charlson comorbidity index has an important role in this cohort despite almost all of the patients were include CCI 2 or 3. Did it mean that the difference in CCI 2 or 3 affected the decision making?
Response) According to our analysis, yes. Most were CCI 2 or 3 points, but there was statistically significant difference between group with CCI≥3 and group with CCI≤2. This results are consistent with the results of a previously published journal in Japan on the long-term outcomes and prognostic factors after ESD for gastric cancer in elderly patients [3].
The authors analyzed the histologic type dividing into two types, differentiated and undifferentiated. How many mixed type EGCs were there?
Response) In the data we analyzed, there were 7 patients classified as mixed type, and these were classified as undifferentiated type and analyzed. The reason why there are few cases classified as mixed thye in our data is presumed that the pathology department of our institution did not place emphasis on the classification of mixed type.
There was at least the patient who was treated within 5 minutes by ESD technique. What was the definition of the procedure time?
Response) It was defined as from the start of dissection for ESD until the dissection is completed and the lesion is separated. We added this information at method section. And this patient was checked again, and it was confirmed that it took about 5 minutes.
DISCUSSION:
The authors described that it may not be reasonable to perform ESD or surgery after non-curative ESD. I would like to agree with this opinion, however, I could not find the data on how many patients treated by additional surgery or rejected additional treatment. It was difficult to understand this sentence without data on additional surgery.
Response) As mentioned in the above answer, 72 patients received additional treatment, and we added this to the Results section.
The authors described that the results of this study showed statistically significant differences in the 4-, 5-, and 10-year OS between the low-risk and high-risk groups (P21 L288).
It was difficult to find the results that showed significant differences.
Response) This is described at the end of “Relationship between OS and clinicopathologic factors” in the Results section. Additionally, Supplementary Figure 2 is a representation of this. In the low-risk group, the 3-, 5-, and 10-year OS rates were 94.6%, 90.1%, and 63.8%, respectively. In the high-risk group, the 3-, 5-, and 10-year OS rates were 80.0%, 62.5%, and 22.1%, respectively. When comparing the two groups in terms of OS, the high-risk group showed a significantly lower OS than the low-risk group (P<0.001).
- Toya, Y.; Endo, M.; Nakamura, S.; Akasaka, R.; Yanai, S.; Kawasaki, K.; Koeda, K.; Eizuka, M.; Fujita, Y.; Uesugi, N. et al. Long-term outcomes and prognostic factors with non-curative endoscopic submucosal dissection for gastric cancer in elderly patients aged ≥ 75 years. Gastric Cancer 2019, 22, 838-844.
